# Eliminating accidental deviations to minimize generalization error and maximize replicability: Applications in connectomics and genomics

Eric W. Bridgeford[1], Shangsi Wang[1], Zeyi Wang[1], Ting Xu[3], Cameron Craddock[3], Jayanta Dey[1], Gregory Kiar[2], William Gray-Roncal[1], Carlo Colantuoni[1], Christopher Douville[1], Stephanie Noble[4], Carey E. Priebe[1], Brian Caffo[1], Michael Milham[3], Xi-Nian Zuo[5], Consortium for Reliability and Reproducibility, Joshua T. Vogelstein[1,6]*

1 Johns Hopkins University, Baltimore, Maryland, United States of America, 2 McGill University, Montreal, Quebec, Canada, 3 Child Mind Institute, New York, New York, United States of America, 4 Yale University, New Haven, Connecticut, United States of America, 5 State Key Laboratory of Cognitive Neuroscience and Learning, International Data Group/McGovern Institute for Brain Research, Beijing Normal University, Beijing, China, 6 Progressive Learning, Baltimore, Maryland, United States of America

* jovo@jhu.edu

**Data Availability Statement:** The data may be found in the following git repository: https://github.

## Abstract

Replicability, the ability to replicate scientific findings, is a prerequisite for scientific discovery and clinical utility. Troublingly, we are in the midst of a replicability crisis. A key to replicability is that multiple measurements of the same item (e.g., experimental sample or clinical participant) under fixed experimental constraints are relatively similar to one another. Thus, statistics that quantify the relative contributions of accidental deviations—such as measurement error—as compared to systematic deviations—such as individual differences—are critical. We demonstrate that existing replicability statistics, such as intra-class correlation coefficient and fingerprinting, fail to adequately differentiate between accidental and systematic deviations in very simple settings. We therefore propose a novel statistic, *discriminability*, which quantifies the degree to which an individual's samples are relatively similar to one another, without restricting the data to be univariate, Gaussian, or even Euclidean. Using this statistic, we introduce the possibility of optimizing experimental design via increasing discriminability and prove that optimizing discriminability improves performance bounds in subsequent inference tasks. In extensive simulated and real datasets (focusing on brain imaging and demonstrating on genomics), only optimizing data *discriminability* improves performance on all subsequent inference tasks for each dataset. We therefore suggest that designing experiments and analyses to optimize discriminability may be a crucial step in solving the replicability crisis, and more generally, mitigating accidental measurement error.

com/neurodata/r-mgc/tree/master/docs/discriminability/paper.

**Funding:** JTV is supported by the National Science Foundation NEURONEX award DMS-1707298, the National Institute of Health award 1R01MH120482-01, the Defense Advanced Research Projects Agency's (DARPA) SIMPLEX program through SPAWAR contract N66001-15-C-4041, and Microsoft Research. X-NZ receives funding supports by the National Basic Research (973) Program (2015CB351702), the Natural Science Foundation of China (81471740, 81220108014), the China - Netherlands CAS-NWO Programme (153111KYSB20160020), Beijing Municipal Science and Tech Commission (Z161100002616023, Z171100000117012), the Start-up Funds for Leading Talents at Beijing Normal University, the Major Project of National Social Science Foundation of China (20&ZD296) and the National Basic Science Data Center "Chinese Data-sharing Warehouse for In-vivo Imaging Brain" (NBSDC-DB-15), and the Guangxi BaGui Scholarship (201621). The funders had no role in study design, data collection and analysis, decision to publish, nor preparation of the manuscript.

**Competing interests:** The authors have declared that no competing interests exist.

## Author summary

In recent decades, the size and complexity of data has grown exponentially. Unfortunately, the increased scale of modern datasets brings many new challenges. At present, we are in the midst of a replicability crisis, in which scientific discoveries fail to *replicate* to new datasets. Difficulties in the measurement procedure and measurement processing pipelines coupled with the influx of complex high-resolution measurements, we believe, are at the core of the replicability crisis. If measurements themselves are not replicable, what hope can we have that we will be able to use the measurements for replicable scientific findings? We introduce the "discriminability" statistic, which quantifies how *discriminable* measurements are from one another, without limitations on the structure of the underlying measurements. We prove that discriminable strategies tend to be strategies which provide better accuracy on downstream scientific questions. We demonstrate the utility of discriminability over competing approaches in this context on two disparate datasets from both neuroimaging and genomics. Together, we believe these results suggest the value of designing experimental protocols and analysis procedures which optimize the discriminability.

This is a *PLOS Computational Biology* Methods paper.

## 1 Introduction

Understanding variability, and the sources thereof, is fundamental to all of data science. Even the first papers on modern statistical methods concerned themselves with distinguishing accidental from systematic variability [1]. Accidental deviations correspond to sources of variance that are not of scientific interest, including measurement noise and artefacts from the particular experiment (often called "batch effects" [2]). Quantifying systematic deviations of the variables of interest, such as variance across items within a study, is often the actual goal of the study. Thus, delineating between these two sources of noise is a central quest in data science, and failure to do so, has been problematic in modern science [3].

Scientific replicability, or the degree to which a result can be replicated using the same methods applied to the same scientific question on new data [4], is key in data science, whether applied to basic discovery or clinical utility [5]. As a rule, if results do not replicate, we can not justifiably trust them [4] (though replication does not imply validation necessarily [6]). The concept of replicability is closely related to the statistical concepts of stability [7] and robustness [5]. Engineering and operations research have been concerned with *reliability* for a long time, as they require that their products are reliable under various conditions. Very recently, the general research community became interested in these issues, as individuals began noticing and publishing failures to replicate across fields, including neuroscience and psychology [8–10].

A number of strategies have been suggested to resolve this "replicability crisis." For example, the editors of "Basic and Applied Social Psychology" have banned the use of p-values [11]. Unfortunately, an analysis of the publications since banning indicates that studies after the ban tended to overstate, rather than understate, their claims, suggesting that this proposal possibly had the opposite effect [12]. More recently, the American Statistical Association released a statement recommending banning the phrase "statistically significant" for similar reasons [13, 14].

A different strategy has been to quantify the repeatability of one's measurements by measuring each sample (or individual) multiple times. Such "test-retest reliability" experiments quantify the relative similarity of multiple measurements of the same item, as compared to different items [15]. Approaches which investigate *measurement repeatability* quantify the degree to which measurements obtained in one session are similar to a set of measurements obtained in a second session, to test replicability [4]. This practice has been particularly popular in brain imaging, where many studies have been devoted to quantifying the repeatability of different univariate properties of the data [16–19]. In practice, however, these approaches have severe limitations. The Intraclass Correlation Coefficient (ICC) is an approach that quantifies the ratio of within item variance to across item variance. The ICC is univariate, with limited applicability to high-dimensional data, and its interpretation suffers from limitations due to its motivating Gaussian assumptions. Previously proposed generalizations of ICC, such as the Image Intraclass Correlation Coefficient (I2C2), generalize ICC to multivariate data, but require large sample sizes to estimate high-dimensional covariance matrices. Further, motivating intuition of I2C2 makes similar Gaussian parametric assumptions as ICC, and therefore exhibits similar limitations. The Fingerprinting Index (Fingerprint) provides a nonparametric and multivariate technique for quantifying test-retest reliability, but its greedy assignment leads it to provide counter-intuitive results in certain contexts. A number of other approaches such as NPAIRS [20, 21] provide general frameworks for evaluating activation-based neuroimaging timeseries experiments, which can be extended to other modalities [22, 23]. A thorough discussion and analysis of these and similar approaches is provided in S1 Text.

Perhaps the most problematic aspect of these approaches is clear from the popular adage, "garbage in, garbage out" [24]. If the measurements themselves are not sufficiently replicable, then scalar summaries of the data cannot be replicable either. This primacy of measurement is fundamental in statistics, so much so that one of the first modern statistics textbook, R.A. Fisher's, "The Design of Experiments" [25], is focused on taking measurements. Motivated by Fisher's work on experimental design, and Spearman's work on measurement, rather than recommending different post-data acquisition inferential techniques, or computing the repeatability of data after collecting, we take a different approach. Specifically, **we advocate for explicitly and specifically designing experiments to ensure that they provide highly replicable data, rather than hoping that they do and performing post-hoc checks after collecting the data**. Thus, we concretely recommend that new studies leverage existing protocols that have previously been established to generate highly replicable data. If no such protocols are available for your question, we recommend designing new protocols in such a way that replicability is explicitly considered (and not compromised) in each step of the design. Experimental design has a rich history, including in psychology [26] and neuroscience [27, 28]. The vast majority of work in experimental design, however, focuses on designing an experiment to answer a particular scientific question. In this big data age, experiments are often designed to answer many questions, including questions not even considered at the time of data acquisition. How can one even conceivably design experiments to obtain data that is particularly useful for those questions?

We propose to design experiments to optimize the *inter-item discriminability* of individual items (for example, participants in a study, or samples in an experiment). This idea is closely inspired by and related to ideas proposed by Cronbach's "Theory of Generalizability" [29, 30]. To do so, we leverage our recently introduced Discr statistic [31]. Discr quantifies the degree to which multiple measurements of the same item are more similar to one another than they are to other items [31], essentially capturing the desiderata of Spearman from over 100 years ago. This statistic has several advantages over existing statistics that one could potentially

use to optimize experimental design. First, it is nonparametric, meaning that its validity and interpretation do not depend on any parametric assumptions, such as Gaussianity. Second, it can readily be applied to multivariate Euclidean data, or even non-Euclidean data (such as images, text, speech, or networks). Third, it can be applied to any stage of the data science pipeline, from data acquisition to data wrangling to data inferences. Finally, and most uniquely, one of the main advantages of `ICC`, is that under certain assumptions, `ICC` can provide an upper bound on predictive accuracy for any subsequent inference task. Specifically, we present here a result generalizing `ICC`'s bound on predictive accuracy to a multivariate additive noise setting. Thus, `Discr` is the *only* non-parametric multivariate measure of test-retest reliability with formal theoretical guarantees of convergence and upper bounds on subsequent inference performance. We show that this property makes `Discr` desirable through empirical simulations and across multiple scientific domains. An important clarification is that high test-retest reliability does not provide any information about the extent to which a measurement coincides with what it is purportedly measuring (construct validity). Even though replicable data are not enough on their own, replicable data are required for stable subsequent inferences.

This manuscript provides the following contributions:

1. Demonstrates that `Discr` is a statistic that adequately quantifies the relative contribution of certain accidental and systematic deviations, whereas previously proposed statistics have not.

2. Formalizes hypothesis tests to assess discriminability of a dataset, and whether one dataset or approach is more discriminable than another. This is in contrast to previously proposed non-parametric approaches to quantify test-retest reliability, that merely provide a test statistic, but no valid test per se.

3. Provides sufficient conditions for `Discr` to provide a lower bound on predictive accuracy. `Discr` is the *only* multivariate measure of replicability that has been theoretically related to criterion validity.

4. Illustrates on 28 neuroimaging datasets from Consortium for Reliability and Reproducibility (CoRR) [32] and two genomics datasets (i) the preprocessing pipelines which maximize `Discr`, and (ii) that maximizing `Discr` is significantly associated with maximizing the amount of information about multiple covariates, in contrast to other related statistics.

5. Provides all source code and data derivatives open access at https://neurodata.io/mgc.

## 2 Methods

### 2.1 The inter-item discriminability statistic

Testing for inter-item discriminability is closely related to, but distinct from, k-sample testing. In k-sample testing we observe k groups, and we want to determine whether they are different *at all*. In inter-item discriminability, the k groups are in fact k different items (or individuals), and we care about whether replicates within each of the k groups are close to each other, which is a specific kind of difference. As a general rule, if one can specify the kind of difference one is looking for, then tests can have more power for that particular kind of difference. The canonical example of this would be an t-test, where if only looks at whether the means are different across the groups, one obtains higher power than if also looking for differences in variances.

To give a concrete example, assume one item has replicates on a circle with radius one, with random angles. Consider another item whose replicates live on another circle, concentric with the first, but with a different radius. The two items differ, and many nonparametric two-

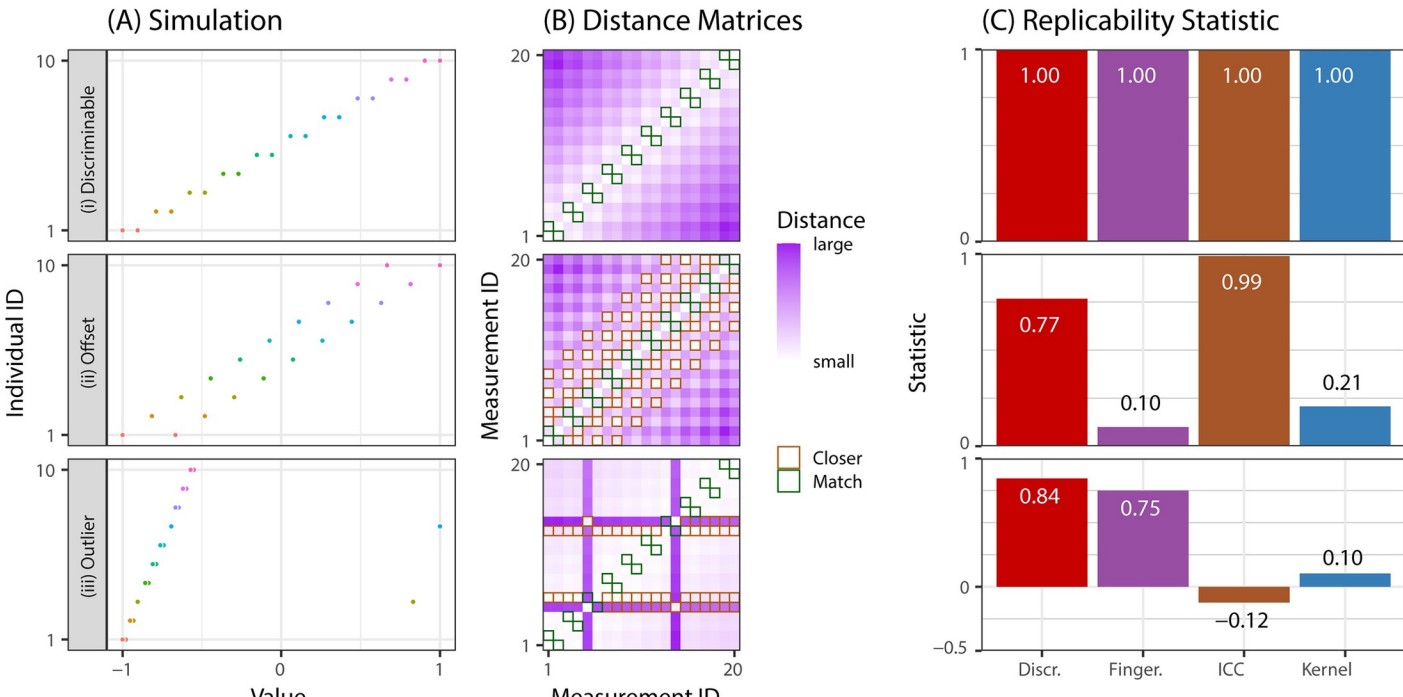

**Fig 1. `Discr` provides a valid discriminability statistic.** Three simulations with characteristic notions of discriminability are constructed with $n = 10$ items each with $s = 2$ measurements. **(A)** The 20 samples, where color indicates the individual associated with a single measurement. **(B)** The distance matrices between pairs of measurements. Samples are organized by item. For each row (measurement), green boxes indicate measurements of the same item, and an orange box indicates a measurement from a different item that is more similar to the measurement than the corresponding measurement from the same item. **(C)** Comparison of four replicability statistics in each simulation. Row *(i)*: Each item is most similar to a repeated measurement from the same item. All discriminability statistics are high. Row *(ii)*: Measurements from the same item are more similar than measurements from different individuals on average, but each item has a measurement from a different item in between. `ICC` is essentially unchanged from *(i)* despite the fact that observations from the same individual are less similar than they were in *(i)*, and both `Fingerprint` and `Kernel` are reduced by about an order of magnitude relative to simulation *(i)*. Row *(iii)*: Two of the ten individuals have an "outlier" measurement, and the simulation is otherwise identical to *(i)*. `ICC` is negative, and `Kernel` provides a small statistic. `Discr` is the only statistic that is robust and valid across all of these simulated examples.

sample tests would indicate so (because one can perfectly identify the item by the radius of the sample). However, the discriminability in this example is not one, because there are samples of either item that are further from other samples of that item than samples from the other item.

On this basis, we developed our inter-item discriminability test statistic (`Discr`), which is inspired by, and builds upon, nonparametric two-sample and k-sample testing approaches called "Energy statistics" [33] and "Kernel mean embeddings" [34] (which are equivalent [35]). These approaches compute all pairwise similarities (or distances) and operate on them. `Discr` differs from these methods in two key ways. First, rather than operating on the magnitudes of all the pairwise distances directly, `Discr` operates on the ranks of the distances, rendering it robust to monotonic transformations of the data [36]. Second, `Discr` only considers comparisons of the ranks of pairwise distances between different items with the ranks of pairwise distances between the same item. All other information is literally discarded, as it does not provide insight into the question of interest.

Fig 1 shows three different simulations illustrating the differences between `Discr` and other replicability statistics, including the fingerprinting index (`Fingerprint`) [37], intraclass correlation coefficient (`ICC`) [38], and `Kernel` [34] (see S1 Text for details). All four statistics operate on the pairwise distance matrices in Fig 1B. However, `Discr`, unlike the other statistics, only considers the elements of each row whose magnitudes are smaller

than the distances within an item. Thus, `Discr` explicitly quantifies the degree to which multiple measurements of the same item are more similar to one another than they are to other items.

**Definition 1** *(Inter-Item Discriminability). Assuming we have n items, where each item has $s_i$ measurements, we obtain $N = \sum_{i=1}^{n} s_i$ total measurements. For simplicity, assume $s_i = 2$ for the definition below, and that there are no ties. Given that, `Discr` can be computed as follows (for a more formal and general definition and pseudocode, please see* S2 Text*):*

1. *Compute the distance between all pairs of samples (resulting in an $N \times N$ matrix),* Fig 1B. *While any measure of distance is permissible, for the purposes of this manuscript, we perform all our experiments using the Euclidean distance.*

2. *Identify replicated measurements of the same individual (green boxes). The number of green boxes is $g = n \times 2$.*

3. *For each measurement, identify measurements that are more similar to it than the other measurement of the same item, i.e., measurements whose magnitude is smaller than that in the green box (orange boxes). Let f be the number of orange boxes.*

4. *Discriminability is defined as fraction of times across-item measurements are smaller than within-item measurements*: $Discr = 1 - \frac{f}{N(N-1)-g}$.

A high `Discr` indicates that within-item measurements tend to be more similar to one another than across-item measurements. See [39] for a theoretical analysis of `Discr` as compared to these and other data replicability statistics. For brevity, we use the term "discriminability" to refer to inter-item discriminability hereafter.

## 2.2 Testing for discriminability

Letting $R$ denote the replicability of a dataset with $n$ items and $s$ measurements per item, and $R_0$ denote the replicability of the same size dataset with zero item specific information, test for replicability is

$$H_0 : R = R_0, \qquad H_A : R > R_0. \tag{1}$$

One can use any 'data replicability' statistic for $R$ and $R_0$ [39]. We devised a permutation test to obtain a distribution of the test statistic under the null, and a corresponding p-value. To evaluate the different procedures, we compute the power of each test, that is, the probability of correctly rejecting the null when it is false (which is one minus type II error; see S5 Text for details).

## 2.3 Testing for better discriminability

Letting $R^{(1)}$ be the replicability of one dataset or approach, and $R^{(2)}$ be the replicability of the second, we have the following comparison hypothesis for replicability:

$$H_0 : R^{(1)} = R^{(2)}, \qquad H_A : R^{(1)} > R^{(2)}. \tag{2}$$

Again, we devised a permutation test to obtain the distribution of the test statistic under the null, and p-values (see S5 Text for details).

## 2.4 Simulation settings

To develop insight into the performance of `Discr`, we consider several different simulation settings (see S4 Text for details). Each setting includes between 2 and 20 items, with 128 total measurements, in two dimensions:

1. **Gaussian** Sixteen items are each distributed according to a spherically symmetric Gaussian, therefore respecting the assumptions that motivate intraclass correlations.

2. **Cross** Two items have Gaussian distributions with the same mean and different diagonal covariance matrices.

3. **Ball/Circle** One item is distributed in the unit ball, the other on the unit circle; Gaussian noise is added to both.

4. **XOR** Each of two items is a mixture of two spherically symmetric Gaussians, but means are organized in an XOR fashion; that is, the means of the first item are (0, 1) and (1, 0), whereas the means of the second are (0, 0) and (1, 1). The implication is that many measurements from a given item are further away than any measurement of the other item.

5. **No Signal** Both items have the same Gaussian distribution.

## 3 Results

### 3.1 Theoretical properties of discriminability

Under reasonably general assumptions, if within-item variability increases, predictive accuracy will decrease. Therefore, a statistic that is sensitive to within-item variance is desirable for optimal experimental design, regardless of the distribution of the data. [40] introduces a univariate parametric framework in which predictive accuracy can be lower-bounded by a decreasing function of `ICC`; as a direct consequence, a strategy with a higher `ICC` will, on average, have higher predictive performance on subsequent inference tasks. Unfortunately, this valuable theoretical result is limited in its applicability, as it is restricted to univariate data, whereas big data analysis strategies often produce multivariate data. We therefore prove the following generalization of this theorem:

**Theorem 1** *Under the multivariate mixture model with the first two moments bounded above, plus additive noise setting, or a sufficient generalization thereof, `Discr` provides a lower bound on the predictive accuracy of a subsequent classification task. Consequently, a strategy with a higher `Discr` provably provides a higher bound on predictive accuracy than a strategy with a lower `Discr`.*

See S3 Text for proof. Correspondingly, this property motivates optimizing experiments to obtain higher `Discr`.

### 3.2 Properties of various replicability statistics

In Fig 1, we highlight the properties of different statistics across a range of basic one-dimensional simulations, all of which display a characteristic notion of replicability: samples of the same item tend to be more similar to one another than samples from different items. In three different univariate simulations we observe two samples from ten items (Fig 1A), and the construct in which replicability statistics will be evaluated:

1. **Discriminable** has each item's samples closer to each other than any other items. The replicability statistic should attain a large value to reflect the high within-item similarity compared to the between-item similarity.

2. **Offset** shifts the second measurement a bit, so that it is further from the first measurement than another item. Replicability statistic should still be high, but lower than the offset simulation.

3. **Outlier** is the same as **discriminable** but includes two items with an outlier measurement. This is another highly reliable setting, so we hope outliers do not significantly reduce the replicability score.

We compare `Discr` to intraclass correlation coefficient (ICC), fingerprinting index (`Fingerprint`) [37], and k-sample kernel testing (`Kernel`) [41] (see S1 Text for details). ICC provides no ability for differentiating between *discriminable* and *offset* simulation, despite the fact that the data in *discriminable* is more replicable than *offset*. While this property may be useful in some contexts, a lack of sensitivity to the offset renders users unable to discern which strategy has a higher test-retest reliability. Moreover, `ICC` is uninterpretable in the case of even a very small number of outliers, where `ICC` is negative. On the other hand, `Fingerprint` suffers from the limitation that if the nearest measurement is anything but a measurement of the same item, it will be at or near zero, as shown in *offset*. `Kernel` also performs poorly in *offset* and in the presence of *outliers*. In contrast, across all simulations, `Discr` shows reasonable construct validity under the given constructs: the statistic is high across all simulations, and highest when repeated measurements of the same item are more similar than measurements from any of the other items.

## 3.3 The power of replicability statistics in multivariate experimental design

We evaluate `Discr`, `PICC` (which applies `ICC` to the top principal component of the data), `I2C2`, `Fingerprint`, and `Kernel` on five two-dimensional simulation settings (see S4 Text for details). Fig 2A shows a two-dimensional scatterplot of each setting, and Fig 2B shows the Euclidean distance matrix between samples, ordered by item.

**3.3.1 Discriminability empirically predicts performance on subsequent inference tasks.** Fig 2C shows the impact of increasing within-item variance on the different simulation settings. The purpose of these simulations is to assess the degree to which `Discr` or the other replicability statistics correspond to downstream predictive accuracy, both under a multivariate Gaussian assumption, and more generally. For the top four simulations, increasing variance decreases predictive accuracy (green line). As desired, `Discr` also decreases nearly perfectly monotonically with decreasing variances. However, only in the first setting, where each item has a spherically symmetric Gaussian distribution, do `I2C2`, `PICC`, and `Fingerprint` drop proportionally. Even in the second (Gaussian) setting, `I2C2`, `PICC`, and `Fingerprint` are effectively uninformative about the within-item variance. And in the third and fourth (non-Gaussian) settings, they are similarly useless. In the fifth simulation they are all at chance levels, as they should be, because there is no information about class in the data. This suggests that of these statistics, only `Discr` and `Kernel` can serve as satisfactory surrogates for predictive accuracy under these quite simple settings.

**3.3.2 A test to determine replicability.** A prerequisite for making item-specific predictions is that items are different from one another in predictable ways, that is, are discriminable. If not, the same assay applied to the same individual on multiple trials could yield unacceptably highly variable results. Thus, prior to embarking on a machine learning search for predictive accuracy, one can simply test whether the data are discriminable at all. If not, predictive accuracy will be hopeless.

Fig 2D shows that `Discr` achieves high power among all competing approaches in all settings and variances. This result demonstrates that despite the fact that `Discr` does not rely on Gaussian assumptions, it still performs nearly as well or better than parametric methods when the data satisfy these assumptions (row (i)). In row (ii) cross, only `Discr` and `Kernel` correctly identify that items differ from one another, despite the fact that the data are Gaussian, though they are not spherically symmetric gaussians. In both rows (iii) ball/disc and (iv) XOR,

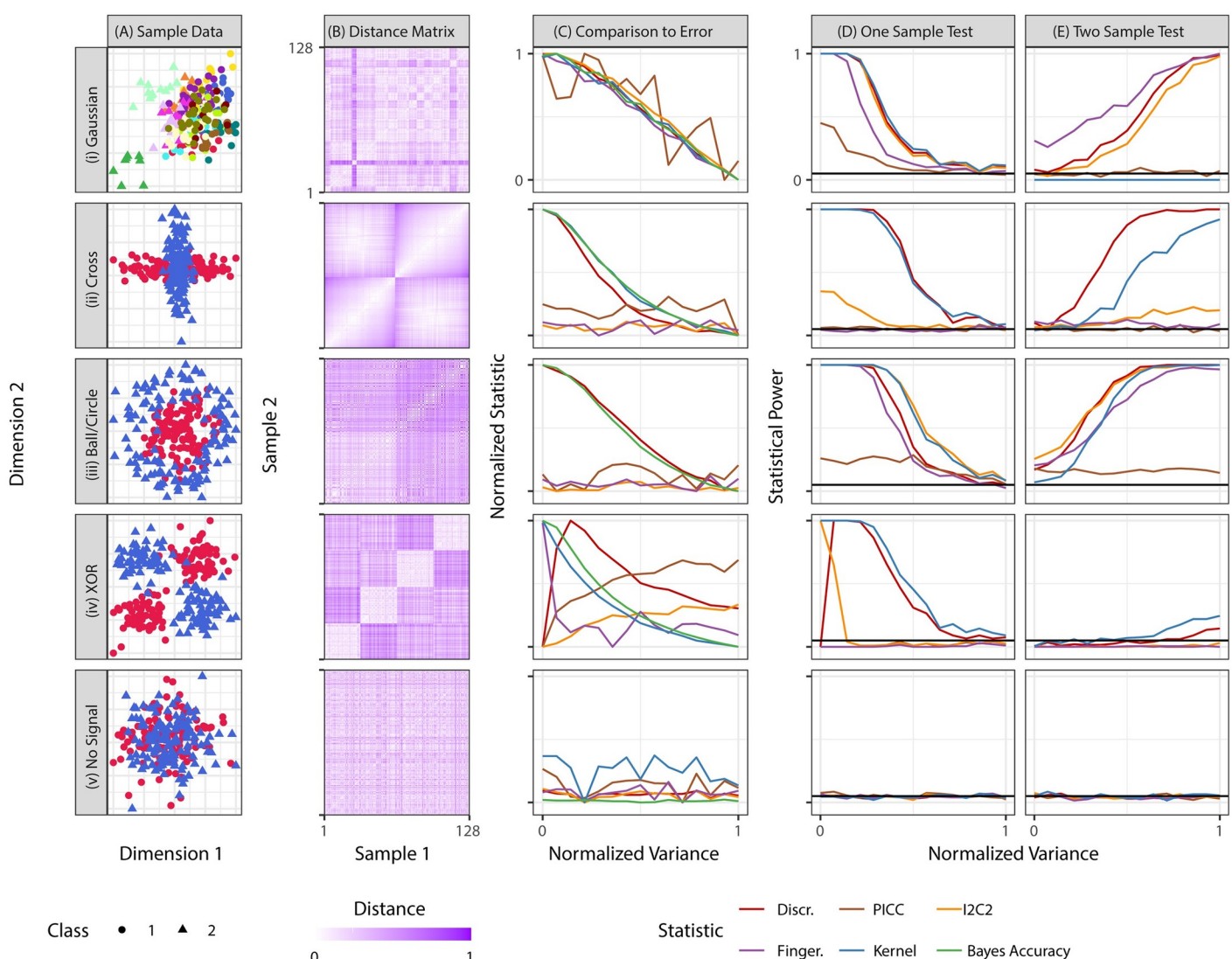

**Fig 2. Multivariate simulations demonstrate the value of optimizing replicability for experimental design.** All simulations are two-dimensional, with 128 samples, with 500 iterations per setting (see S4 Text for details). **(A)** For each setting, class label is indicated by shape, and color indicates item identity. **(B)** Euclidean distance matrix between samples within each simulation setting. Samples are organized by item. Simulation settings in which items are discriminable tend to have a block structure where samples from the same item are relatively similar to one another. **(C)** Replicability statistic versus variance. Here, we can compute the Bayes accuracy (the best one could perform to predict class label) as a function of variance. `Discr` and `Kernel` are mostly monotonic relative to within-item variance across all settings, suggesting that one can predict improved performance via improved `Discr`. **(D)** Test of whether data are discriminable. `Discr` typically achieves high power among the alternative statistics in all cases. **(E)** Comparison test of which approach is more discriminable. `Discr` is the only statistic which achieves high power in all settings in which any statistic was able to achieve high power.

most statistics perform well despite the non-Gaussianity of the data. And when there is no signal, all tests are valid, achieving power less than or equal to the critical value. Non-parametric `Discr` therefore has the power of parametric approaches for data at which those assumptions are appropriate, and much higher power for other data. `Kernel` performs comparably to `Discr` in these settings.

**3.3.3 A test to compare replicabilities.** Given two experimental designs—which can differ either by acquisition and/or analysis details—are the measurements produced by one method more discriminable than the other? Fig 2D shows `Discr` typically achieves the

highest power among all statistics considered. Specifically, only `Fingerprint` achieves higher power in the Gaussian setting, but it achieves almost no power in the cross setting. `Kernel` achieves comparably lower power for most settings and no power for the Gaussian, as does `PICC`. `I2C2` achieves similar power to `Discr` only for the Gaussian and ball/disc setting. All tests are valid in that they achieve a power approximately equal to or below the critical value when there is no signal. Note that these comparisons are not the typical "k-sample comparisons" with many theoretical results, rather, they are comparing across multiple disparate k-sample settings. Thus, in general, there is a lack of theoretical guarantees for this setting. Nonetheless, the fact that `Discr` achieves nearly equal or higher power than the statistics that build upon Gaussian methods, even under Gaussian assumptions, suggests that `Discr` will be a superior metric for optimal experimental design in real data.

### 3.4 Optimizing experimental design via maximizing replicability in human brain imaging data

**3.4.1 Human brain imaging data acquisition and analysis.**   Consortium for Reliability and Reproducibility (CoRR) [42] has generated functional, anatomical, and diffusion magnetic resonance imaging (dMRI) scans from >1,600 participants, often with multiple measurements, collected through 28 different datasets (22 of which have both age and sex annotation) spanning over 20 sites. Each of the sites use different scanners, technicians, scanning protocols, and retest follow up procedures, thereby representing a wide variety of different acquisition settings with which one can test different analysis pipelines. S6 Text protocol metadata associated with each individual dataset. Fig 3A shows the six stage sequence of analysis steps for converting the raw fMRI data into networks or connectomes, that is, estimates of the strength of connections between all pairs of brain regions. At each stage of the pipeline, we consider several different "standard" approaches, that is, approaches that have previously been proposed in the literature, typically with hundreds or thousands of citations [43]. Moreover, they have all been collected into an analysis engine, called Configurable Pipeline for the Analysis of Connectomes (C-PAC) [44]. In total, for the six stages together, we consider $2 \times 2 \times 2 \times 2 \times 4 \times 3 = 192$ different analysis pipelines. Because each stage is nonlinear, it is possible that the best sequence of choices is not equivalent to the best choices on their own. For this reason, publications that evaluate a given stage using any metric, could result in misleading conclusions if one is searching for the best sequence of steps [45]. The dMRI connectomes were acquired via 48 analysis pipelines using the Neurodata MRI Graphs (`ndmg`) pipeline [46]. S6 Text provides specific details for both fMRI and dMRI analysis, as well as the options attempted.

**3.4.2 Different analysis strategies yield widely disparate stabilities.**   The analysis strategy has a large impact on the `Discr` of the resulting fMRI connectomes (Fig 3B). Each column shows one of 64 different analysis strategies, ordered by how significantly different they are from the pipeline with highest `Discr` (averaged over all datasets, tested using the above comparison test). Interestingly, pipelines with worse average `Discr` also tend to have higher variance across datasets. The best pipeline, FNNNCP, uses FSL registration, no frequency filtering, no scrubbing, no global signal regression, CC200 parcellation, and converts edges weights to ranks. While all strategies across all datasets with multiple participants are significantly discriminable at $\alpha = 0.05$ (`Discr` goodness of fit test), the majority of the strategies ($51/64 \approx$ 80%) show significantly worse `Discr` than the optimal strategy at $\alpha = 0.05$ (`Discr` comparison test).

**3.4.3 Discriminability identifies which acquisition and analysis decisions are most important for improving performance.**   While the above analysis provides evidence for

## (A) Processing Strategies Evaluated

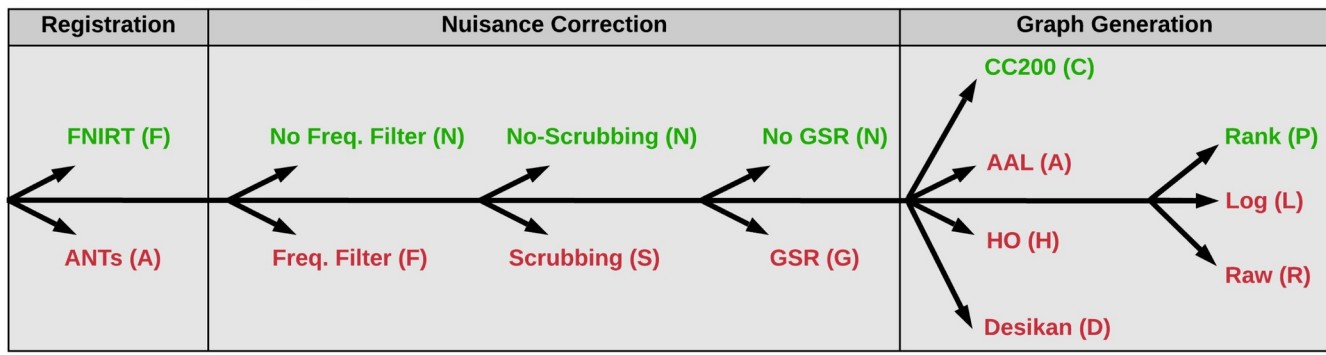

## (B) Comparing Discriminability Across 64 Preprocessing Strategies

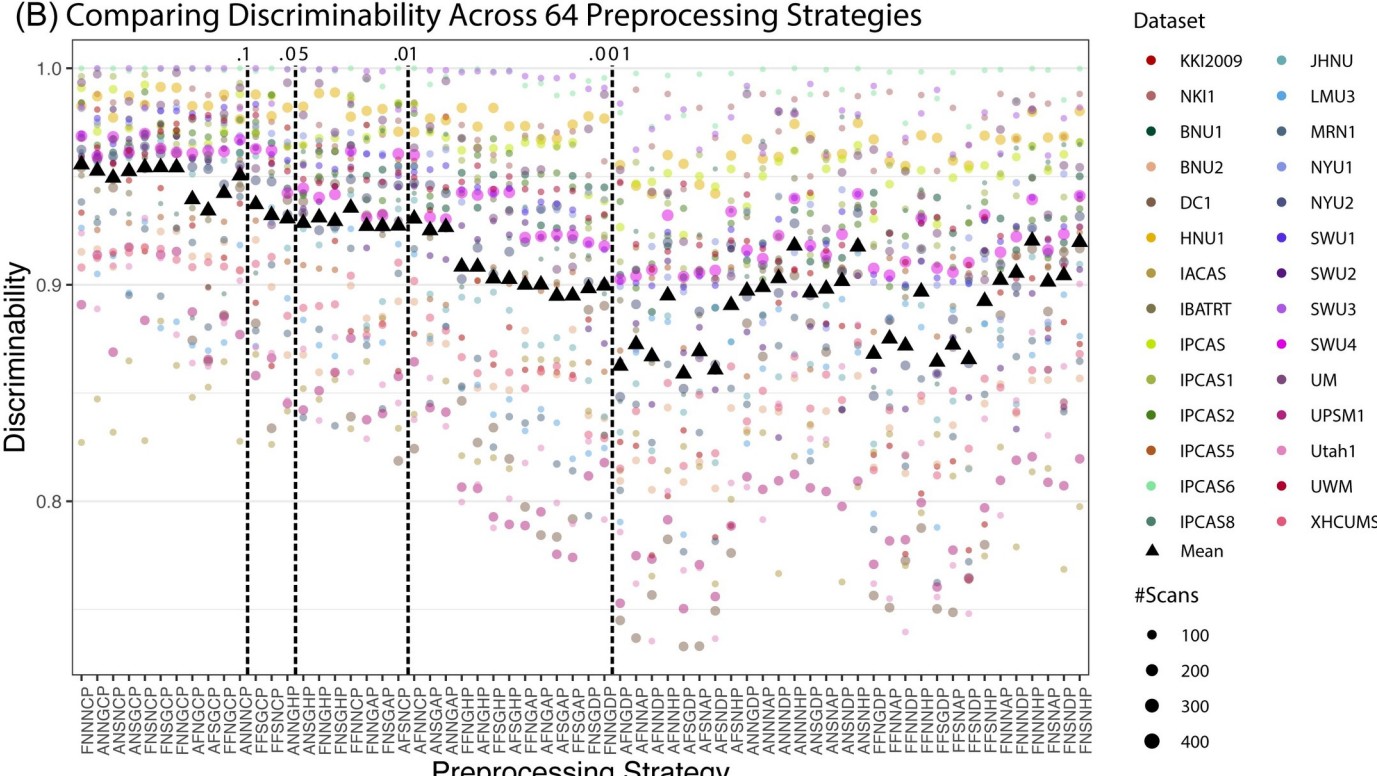

**Fig 3. Different analysis strategies yield widely disparate stabilities. (A)** Illustration of analysis options for the 192 fMRI pipelines under consideration (described in S6 Text). The sequence of options corresponding to the best performing pipeline overall are in green. **(B)** `Discr` of fMRI Connectomes analyzed using 64 different pipelines. Functional correlation matrices are estimated from 28 multi-session studies from the CoRR dataset using each pipeline. The analysis strategy codes are assigned sequentially according to the abbreviations listed for each step in **(A)**. The mean `Discr` per pipeline is a weighted sum of its stabilities across datasets. Each pipeline is compared to the optimal pipeline with the highest mean `Discr`, FNNNCP, using the above comparison hypothesis test. The remaining strategies are arranged according to *p*-value, indicated in the top row.

which *sequence* of analysis steps is best, it does not provide information about which choices individually have the largest impact on overall `Discr`. To do so, it is inadequate to simply fix a pipeline and only swap out algorithms for a single stage, as such an analysis will only provide information about that fixed pipeline. Therefore, we evaluate each choice in the context of all 192 considered pipelines in Fig 4A. The pipeline constructed by identifying the best option for each analysis stage is FNNGCP (Fig 4A). Although it is not exactly the same as the pipeline with

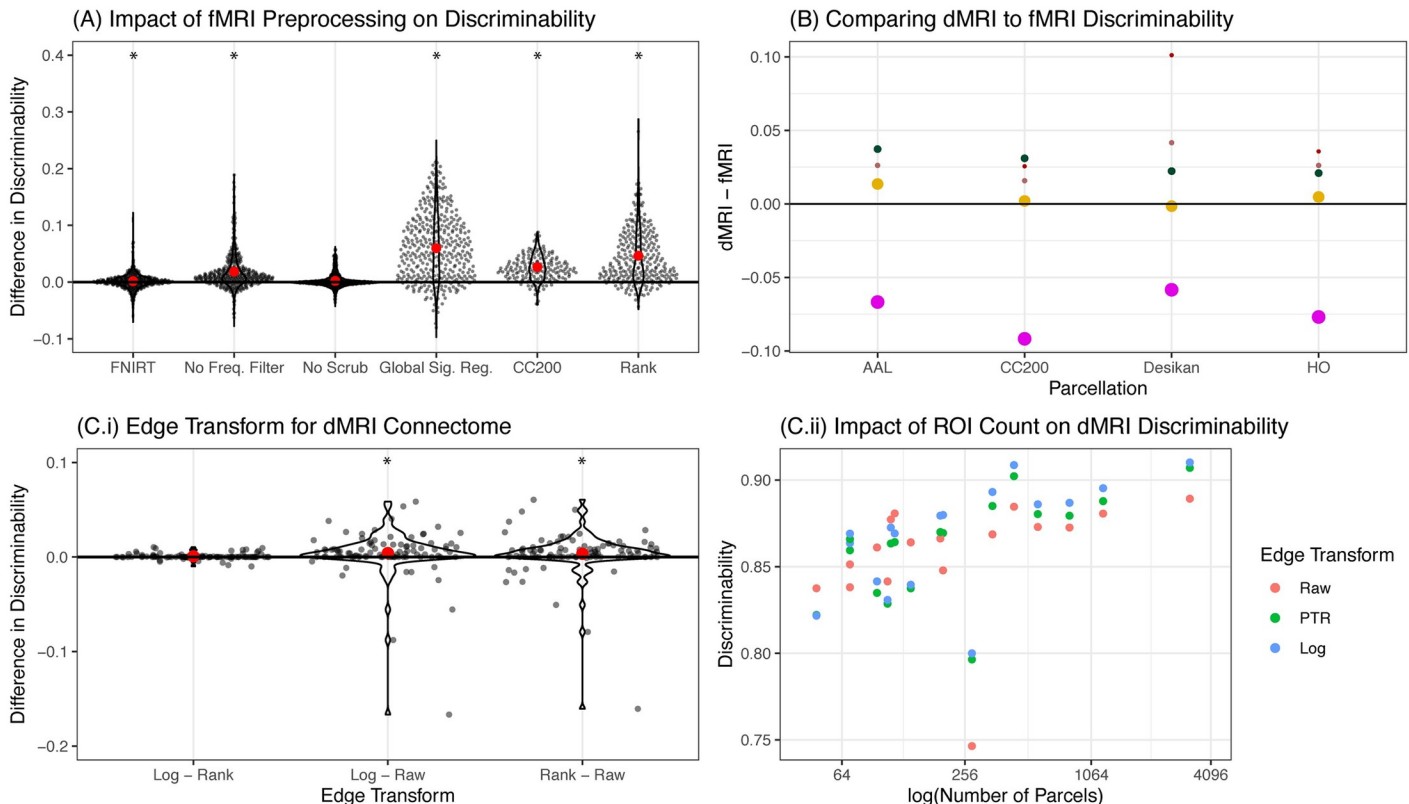

**Fig 4. Parsing the relative impact on `Discr` of various acquisition and analytic choices. (A)** The pipelines are aggregated for a particular analysis step, with pairwise comparisons with the remaining analysis options held fixed. The beeswarm plot shows the difference between the overall best performing option and the second best option for each stage (mean in red) with other options held equal; the *x*-axis label indicates the best performing strategy. The best strategies are FNIRT, no frequency filtering, no scrubbing, global signal regression, the CC200 parcellation, and ranks edge transformation. A Wilcoxon signed-rank test is used to determine whether the mean for the best strategy exceeds the second best strategy: a * indicates that the *p*-value is at most 0.001 after Bonferroni correction. Of the best options, only no scrubbing is *not* significantly better than alternative strategies. Note that the options that perform marginally the best are not significantly different than the best performing strategy overall, as shown in Fig 3. **(B)** A comparison of the stabilities for the 4 datasets with both fMRI and dMRI connectomes. dMRI connectomes tend to be more discriminable, in 14 of 20 total comparisons. Color and point size correspond to the study and number of scans, respectively (see Fig 3B). **(C.i)** Comparing raw edge weights (Raw), ranking (Rank), and log-transforming the edge-weights (Log) for the diffusion connectomes, the Log and Rank transformed edge-weights tend to show higher `Discr` than Raw. **(C.ii)** As the number of ROIs increases, the `Discr` tends to increase.

highest `Discr` (FNNNCP), it is also not much worse (`Discr` 2-sample test, p-value $\approx 0.14$). Moreover, except for scrubbing, each stage has a significant impact on `Discr` after correction for multiple hypotheses (Wilcoxon signed-rank statistic, *p*-values all $< 0.001$).

Another choice is whether to estimate connectomes using functional or diffusion MRI (Fig 4B). Whereas both data acquisition strategies have known problems [47], the `Discr` of the two experimental modalities has not been directly compared. Using four datasets from CoRR that acquired both fMRI and dMRI on the same subjects, and have quite similar demographic profiles, we tested whether fMRI or dMRI derived connectomes were more discriminable. The pipelines being considered were the best-performing fMRI pre-processing pipeline (FNNNCP) against the dMRI pipeline with the CC200 parcellation. For three of the four datasets, dMRI connectomes were more discriminable. This is not particularly surprising, given the susceptibility of fMRI data to changes in state rather than trait (e.g., amount of caffeine prior to scan [44]).

The above results motivate investigating which aspects of the dMRI analysis strategy were most effective. We focus on two criteria: how to scale the weights of connections, and how

many regions of interest (ROIs) to use. For scaling the weights of the connections, we consider three possible criteria: using the raw edge-weights ("Raw"), taking the log of the edge-weights ("Log"), and ranking the non-zero edge weights in sequentially increasing order ("Rank"). Fig 4C.i shows that both rank and log transform significantly exceed raw edge weights (Wilcoxon signed-rank statistic, sample size = 60, p-values all < 0.001). Fig 4C.ii shows that parcellations with larger numbers of ROIs tend to have higher `Discr`. Unfortunately, most parcellations with semantic labels (e.g., visual cortex) have hundreds not thousands of parcels. This result therefore motivates the development of more refined semantic labels.

**3.4.4 Optimizing discriminability improves downstream inference performance.** We next examined the relationship between the `Discr` of each pipeline, and the amount of information it preserves about two properties of interest: sex and age. Based on the simulations above, we expect that analysis pipelines with higher `Discr` will yield connectomes with more information about covariates. Indeed, Fig 5 shows that, for virtually every single dataset including sex and age annotation (22 in total), a pipeline with higher `Discr` tends to preserve more information about both covariates. The amount of information is quantified by the effect size of the distance correlation `DCorr` (which is exactly equivalent to `Kernel` [36, 48]), a statistic that quantifies the magnitude of association for both linear and nonlinear dependence structures. In contrast, if one were to use either `Kernel` or `I2C2` to select the optimal pipeline, for many datasets, subsequent predictive performance would degrade. `Fingerprint` performs similarly to `Discr`, while `PICC` provides a slight decrease in performance on this dataset. These results are highly statistically significant: the slopes of effect size versus `Discr` and `Fingerprint` across datasets are significantly positive for both age and sex in 82 and 95 percent of all studies, respectively (robust $Z$-test, $\alpha = 0.05$). `Kernel` performs poorly, basically always, because $k$-sample tests are designed to perform well with many samples from a small number of different populations, and questions of replicability across repeated measurements have a few samples across many different populations.

## 3.5 Replicability of genomics data

The first genomics study aimed to explore variation in gene expression across human induced pluripotent stem cell (hiPSC) lines with between one and seven replicates [49]. This data includes RNAseq data from 101 healthy individuals, comprising 38 males and 63 females. Expression was interrogated across donors by studying up to seven replicated iPSC lines from each donor, yielding bulk RNAseq data from a total of 317 individual hiPSC lines. While the pipeline includes many steps, we focus here for simplicity on (1) counting, and (2) normalizing. The two counting approaches we study are the raw hiPSC lines and the count-per-million (CPM). Given counts, we consider three different normalization options: Raw, Rank, and Log-transformed (as described above). The task of interest was to identify the sex of the individual.

The second genomics study [50] includes 331 individuals, consisting of 135 patients with non-metastatic cancer and 196 healthy controls, each with eight DNA samples. The study leverages a PCR-based assay called Repetitive element aneuploidy sequencing system to analyze ~750,000 amplicons distributed throughout the genome to investigate the presence of aneuploidy (abnormal chromosome counts) in samples from cancer patients (see S6 Text for more details). The possible processing strategies include using the raw amplicons or the amplicons downsampled by a factor of $5 \times 10^5$ bases, $5 \times 10^6$ bases, $5 \times 10^7$ bases, or to the individual chromosome level (the *resolution* of the data), followed by normalizing through the previously described approaches (Raw, Rank, Log-transformed) yielding $5 \times 3 = 15$ possible strategies in

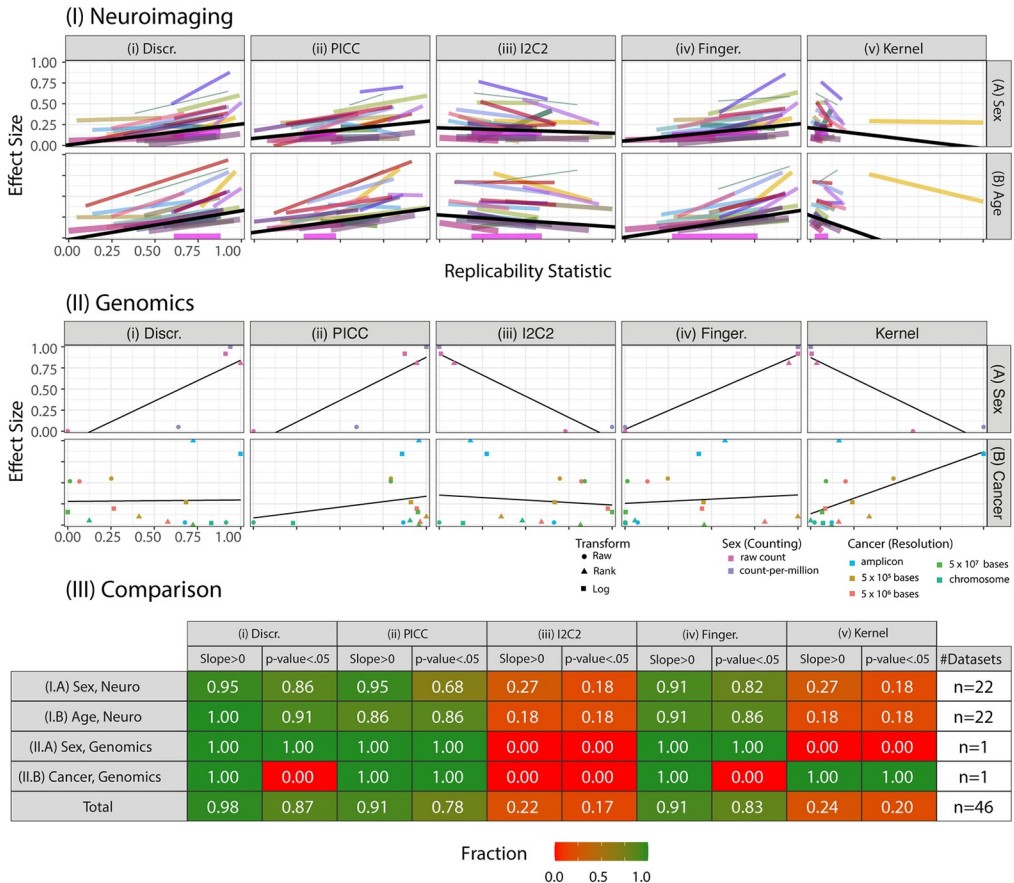

**Fig 5. Optimizing `Discr` improves downstream inference performance.** Using the connectomes from the 64 pipelines with raw edge-weights, we examine the relationship between connectomes vs sex and age. The columns evaluate difference approaches for computing pipeline effectiveness, including **(i)** `Discr`, **(ii)** `PICC`, **(iii)** Average Fingerprint Index `Fingerprint`, **(iv)** `I2C2`, and **(v)** `Kernel`. Each panel shows reference pipeline replicability estimate (*x-axis*) versus effect size of the association between the data and the sex, age, or cancer status of the individual as measured by `DCorr` (*y-axis*). Both the *x* and *y* axes are normalized by the minimum and maximum statistic. These data are summarized by a single line per study, which is the regression of the normalized effect size onto the normalized replicability estimate as quantified by the indicated reference statistic. **(I)** The results for the neuroimaging data, as described in Section 3.4. Color and line width correspond to the study and number of scans, respectively (see Fig 3B). The solid black line is the weighted mean over all studies. `Discr` is the only statistic in which *nearly all* slopes are positive. Moreover, the corrected *p*-value [51, 52] is significant across most datasets for both covariates ($\frac{39}{44} \approx .89$ *p*-values < .001). This indicates that pipelines with higher `Discr` correspond to larger effect sizes for the covariate of interest, and that this relationship is stronger for `Discr` than other statistics. A similar experiment is performed on two genomics datasets, measuring the effects due to sex and whether an individual has cancer. **(III)** indicates the fraction of datasets with positive slopes and with significantly positive slopes, ranging from 0 ("None", red) to 1 ("All", green), at both the task and aggregate level. `Discr` is the statistic where the most datasets have positive slopes, and the statistic where the most datasets have significantly positive slopes, across the neuroimaging and genomics datasets considered. S6 Text details the methodologies employed.

total. The task of interest was to identify whether the sample was collected from a cancer patient or a healthy control.

Across both tasks, slope for discriminability is positive, and for the first task, the slope is significantly bigger than zero (robust *Z*-test, *p*-value = .001, $\alpha$ = .05). `Fingerprint` and `Kernel` are similarly only informative for one of the two genomics studies. For `PICC`, in both datasets the slope is positive and the effect is significant. `I2C2` does not provide value for subsequent inference.

## 4 Discussion

We propose the use of the `Discr` statistic as a simple and intuitive measure for experimental design featuring multiple measurements. Numerous efforts have established the value of *quantifying* repeatability and replicability (or discriminability) using parametric measures such as `ICC` and `I2C2`. However, they have not been used to optimize replicability—that is, they are only used post-hoc to determine replicability, not used as criteria for searching over the design space—nor have non-parametric multivariate generalizations of these statistics been available. We derive goodness of fit and comparison (equality) tests for `Discr`, and demonstrate via theory and simulation that `Discr` provides numerous advantages over existing techniques across a range of simulated settings. Our neuroimaging and genomics use-cases exemplify the utility of these features of the `Discr` framework for optimal experimental design.

An important consideration is that quantifying test-retest reliability and replicability with multiple measurements may seem like a limitation for many fields, in which the end derivative typically used for inference may be just a single sample for each item measured. However, a single measurement may often consist of many sub-measurements for a single individual, each of which are combined to produce the single derivative work. For example in brain imaging, a functional Magnetic Resonance Imaging (fMRI) scan consists of tens to thousands of scans of the brain at numerous time points. In this case, the image can be broken into identical-width time windows to coerce a dataset in which discriminability can be investigated. In another example taken directly from the cancer genomics experiment below, a genomics count table was produced from eight independent experiments, each of which yielded a single count table. The last step of their pre-processing procedure was to aggregate to produce the single summary derivative that the experimenters traditionally considered a single measurement. In each case, the typical "measurement" unit can really be thought of as an aggregate of multiple smaller measurement units, and a researcher can leverage these smaller measurements as a surrogate for multiple measurements. In the neuroimaging example, the fMRI scan can be segmented into identical-width sub-scans with each treated as a single measurement, and in the genomics example, the independent experiments can each be used as a single measurement.

`Discr` provides a number of connections with related statistical algorithms worth further consideration. `Discr` is related to energy statistics [53], in which the statistic is a function of distances between observations [33]. Energy statistics provide approaches for goodness-of-fit (one-sample) and equality testing (two-sample), and multi-sample testing [54]. However, we note an important distinction: a goodness of fit test for discriminability can be thought of as a *K*-sample test in the classical literature, and a comparison of discriminabilities is analogous to a comparison of *K*-sample tests. Further, similar to `Discr`, energy statistics make relatively few assumptions. However, energy statistics requires a large number of measurements per item, which is often unsuitable for biological data where we frequently have only a small number of repeated measurements. `Discr` is most closely related to multiscale generalized correlation (`MGC`) [36, 48], which combines energy statistics with nearest neighbors, as does `Discr`. Like many energy-based statistics, `Discr` relies upon the construction of a distance matrix. As such, `Discr` generalizes readily to high-dimensional data, and many packages accelerate distance computation in high-dimensionals [55].

### Limitations

While `Discr` provides experimental design guidance for big data, other considerations may play a role in a final determination of the practical utility of an experimental design. For example, the connectomes analyzed here are *resting-state*, as opposed to *task-based* fMRI connectomes. Recent literature suggests that the global signal in a rs-fMRI scan may be correlated

heavily with signals of interest for task-based approaches [56, 57], and therefore removal may be inadvisable. Thus, while `Discr` is an effective tool for experimental design, knowledge of the techniques in conjunction with the constructs under which successive inference will be performed remains essential. Further, in this study, we only consider the Euclidean distance, which may not be appropriate for all datasets of interest. For example, if the measurements live in a manifold (such as images, text, speech, and networks), one may be interested in dissimilarity or similarity functions other than Euclidean distance. To this end, `Discr` readily generalizes to alternative comparison functions, and will produce an informative result as long as the choice of comparison function is appropriate for the measurements.

It is important to emphasize that `Discr`, as well the related statistics, are neither necessary, nor sufficient, for a measurement to be practically useful. For example, categorical covariates, such as sex, are often meaningful in an analysis, but not discriminable. Human fingerprints are discriminable, but typically not biologically useful. In this sense, while discriminability provides a valuable link between test-retest reliability and criterion validity for multivariate data, one must be careful to consider other notions of validity prior to the selection of a measurement. In addition, none of the statistics studied here are immune to sample characteristics, thus interpreting results across studies deserves careful scrutiny. For example, having a sample with variable ages will increase the inter-subject dissimilarity of any metric dependent on age (such as the connectome). Additionally, discriminability can be decomposed into within and between-class discriminabilities, so that class-specific effects may be examined in isolation, as described in S7 Text. Future work could explore how these two quantities may be incorporated into the experimental design procedure.

Moreover, if multiple strategies are saturated at a perfect discriminability (`Discr` = 1), it does not provide an informative way to differentiate between these strategies. One could trivially augment the discriminability procedure to compare within-item distances to a scaled and/or shifted transformation of between-item distances, thereby rendering perfect discriminability arbitrarily difficult. With these caveats in mind, `Discr` remains a key experimental design consideration across a wide variety of settings.

## Conclusion

The use-cases provided herein serve to illustrate how `Discr` can be used to facilitate experimental design, and mitigate replicability issues. We envision that `Discr` will find substantial applicability across disciplines and sectors beyond brain imaging and genomics, such pharmaceutical research. To this end, we provide open-source implementations of `Discr` for both `Python` and R [58, 59]. Code for reproducing all the figures in this manuscript is available at https://neurodata.io/mgc.

## Supporting information

**S1 Text. Background information on repeatability statistics.**
(PDF)

**S2 Text. Population and sample discriminability.**
(PDF)

**S3 Text. Theoretical bound for downstream inference.**
(PDF)

**S4 Text. Simulation settings.**
(PDF)

**S5 Text. Hypothesis testing.**
(PDF)

**S6 Text. Data descriptions and details for real data analysis.**
(PDF)

**S7 Text. Extensions of discriminability.**
(PDF)

## Acknowledgments

The authors would like to thank Iris Van Rooij and the Neurodata team for their valuable feedback on this manuscript.

## Author Contributions

**Conceptualization:** Eric W. Bridgeford, Shangsi Wang, Cameron Craddock, Carey E. Priebe, Michael Milham, Joshua T. Vogelstein.

**Data curation:** Eric W. Bridgeford, Cameron Craddock, Gregory Kiar, William Gray-Roncal, Carlo Colantuoni, Christopher Douville, Michael Milham, Xi-Nian Zuo, Joshua T. Vogelstein.

**Formal analysis:** Eric W. Bridgeford, Shangsi Wang, Brian Caffo.

**Investigation:** Eric W. Bridgeford, Joshua T. Vogelstein.

**Methodology:** Eric W. Bridgeford, Shangsi Wang, Zeyi Wang, Cameron Craddock, Carey E. Priebe, Brian Caffo, Michael Milham, Joshua T. Vogelstein.

**Software:** Eric W. Bridgeford, Cameron Craddock, Jayanta Dey.

**Supervision:** Carey E. Priebe, Brian Caffo, Michael Milham, Joshua T. Vogelstein.

**Validation:** Eric W. Bridgeford, Ting Xu, Gregory Kiar, Joshua T. Vogelstein.

**Visualization:** Eric W. Bridgeford.

**Writing – original draft:** Eric W. Bridgeford, Joshua T. Vogelstein.

**Writing – review & editing:** Eric W. Bridgeford, Zeyi Wang, Ting Xu, Cameron Craddock, Gregory Kiar, Carlo Colantuoni, Christopher Douville, Stephanie Noble, Brian Caffo, Michael Milham, Joshua T. Vogelstein.

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
