## [Decision Letter · Decision Letter 0]

28 Apr 2021

Dear Dr. Vogelstein,

Thank you very much for submitting your manuscript "Eliminating accidental deviations to minimize generalization error and maximize reliability: applications in connectomics and genomics" for consideration at PLOS Computational Biology. As with all papers reviewed by the journal, your manuscript was reviewed by members of the editorial board and by several independent reviewers. The reviewers appreciated the attention to an important topic. Based on the reviews, we are likely to accept this manuscript for publication, providing that you modify the manuscript according to the review recommendations.

Sincerely,

Blake A Richards

Associate Editor

PLOS Computational Biology

Jian Ma

Deputy Editor

PLOS Computational Biology

[LINK]

Reviewer's Responses to Questions

**Comments to the Authors:**

Reviewer #1: My review is uploaded as attachment file.

Reviewer #3: Spurred by the reproducibility issue, and motivated by classical works in statistics (Spearman, Fisher), the authors expand on a statistic called discriminability, which quantifies the degree to which an individual’s samples are similar to one another. (Some of the authors proposed this index earlier.) The aim is to minimize generalization error and maximize reliability. Discriminability is a nonparametric index that is appropriate for a wide range of data types (text, images). By using simulations, proving theorems (for additive noise), and demonstrating its use on neuroimaging and genomic data, they show that their index outperforms intraclass correlation (ICC), fingerprinting index, and kernel methods.

Comment.

The authors say that the ICC’s Gaussian roots are a problem. However, their theorems deal with Gaussian errors (the mixture of Gaussians seems fine, for they are flexible). What happens if the errors are not Gaussian ? Many homogeneous populations are skewed, so skew-normal or related distributions may be more appropriate.

The paper is very well written; I spotted a few awkward phrasings

p 4

why SUM i * s_{i} ? why not SUM s_{i} ?

indentify —> identify

for the below definition —> for the definition below

p 24

to overcome to requirement of ICC —> to overcome the requirement of ICC

**Have all data underlying the figures and results presented in the manuscript been provided?**

Reviewer #1: Yes

PLOS authors have the option to publish the peer review history of their article (what does this mean?). If published, this will include your full peer review and any attached files.

Reviewer #1: **Yes: **Stephen C. Strother

Reviewer #3: No

**Have the authors made all data and (if applicable) computational code underlying the findings in their manuscript fully available?**

Reviewer #3: Yes

Figure Files:

Data Requirements:

Reproducibility:

References:

---

## [Decision Letter · Decision Letter 1]

14 Jul 2021

Dear Dr. Vogelstein,

We are pleased to inform you that your manuscript 'Eliminating accidental deviations to minimize generalization error and maximize replicability: applications in connectomics and genomics' has been provisionally accepted for publication in PLOS Computational Biology.

Before your manuscript can be formally accepted you will need to complete some formatting changes, which you will receive in a follow up email. A member of our team will be in touch with a set of requests. Please also be sure when producing the final version of the paper for publication to address the remaining few comments from Reviewer 1, particularly the note regarding NPAIRS.

Best regards,

Blake A Richards

Associate Editor

PLOS Computational Biology

Jian Ma

Deputy Editor

PLOS Computational Biology

Reviewer's Responses to Questions

**Comments to the Authors:**

Reviewer #1: Response to Rev1, Comment 3: I appreciate the authors expanding their list of prior work to include NPAIRS. This is a minor issue that should not distract from their contributions in the rest of the manuscript, but NPAIRS is and has been applied without SPMs beyond the Neuroimaging domain. One only needs to modify the similarity distance metric as shown using mutual information in the original companion paper with sensitivity maps (Kjems et al., 2002: https://pubmed.ncbi.nlm.nih.gov/11906219/) and then illustrated with Raman spectroscopy (Sigurdsson et al., 2004: https://pubmed.ncbi.nlm.nih.gov/15490825/), which is well beyond the Neuroimaging domain. Furthermore, NPAIRS has also been modified to apply to the proteomics domain (https://pubmed.ncbi.nlm.nih.gov/20531955/) and application to images of GWAS SNP expression arrays is relatively trivial. Finally, NPAIRS can readily also be applied in connectomics by using a graph similarity metric and some form of sensitivity analysis on graph ensembles, although to my knowledge this has not been published.

I think the authors important contributions in their manuscript will stand on their own right without unnecessarily claiming limitations of other approaches that are easily refuted by the literature.

Response to Rev 1, Comment 9: I think the English you want is: Illustrates […] (i) the preprocessing pipelines that maximise … . You could also use (i) which preprocessing pipelines maximise … . The first is better Strunk and White English but the second also captures the same meaning.

Response to Rev 1, Comment 14: Supp. S7 is an interesting and useful addition to the paper. I look forward to further work from this group following these ideas.

Reviewer #3: The authors have addressed my concerns.

**Have the authors made all data and (if applicable) computational code underlying the findings in their manuscript fully available?**

Reviewer #1: Yes

Reviewer #3: Yes

PLOS authors have the option to publish the peer review history of their article (what does this mean?). If published, this will include your full peer review and any attached files.

Reviewer #1: **Yes: **Stephen C. Strother

Reviewer #3: No

---

## [Editor Report · Acceptance letter]

7 Sep 2021

PCOMPBIOL-D-21-00089R1 

Eliminating accidental deviations to minimize generalization error and maximize replicability: applications in connectomics and genomics

Dear Dr Vogelstein,

I am pleased to inform you that your manuscript has been formally accepted for publication in PLOS Computational Biology. Your manuscript is now with our production department and you will be notified of the publication date in due course.

With kind regards,

Zsofi Zombor
